# Uniformity of Food Protein Interpretation Amongst Dietitians for Patients with Phenylketonuria (PKU): 2020 UK National Consensus Statements

**DOI:** 10.3390/nu12082205

**Published:** 2020-07-24

**Authors:** Sharon Evans, Sarah Adam, Sandra Adams, Heather Allen, Catherine Ashmore, Sarah Bailey, Janette Banks, Harriet Churchill, Barbara Cochrane, Jennifer Cook, Clare Dale, Anne Daly, Marjorie Dixon, Carolyn Dunlop, Charlotte Ellerton, Anita Emm, Sarah Firman, Suzanne Ford, Moira French, Joanna Gribben, Anne Grimsley, Ide Herlihy, Melanie Hill, Shirley Judd, Karen Lang, Jo Males, Joy McDonald, Nicola McStravick, Chloe Millington, Camille Newby, Catharine Noble, Rachel Pereira, Alex Pinto, Louise Robertson, Abigail Robotham, Kathleen Ross, Kath Singleton, Rachel Skeath, Allyson Terry, Karen Van Wyk, Fiona White, Lucy White, Jo Wildgoose, Alison Woodall, Anita MacDonald

**Affiliations:** 1Birmingham Women’s & Children’s NHS Foundation Trust, Birmingham B4 6NH, UK; catherine.ashmore@nhs.net (C.A.); a.daly3@nhs.net (A.D.); alex.pinto@nhs.net (A.P.); anita.macdonald@nhs.net (A.M.); 2Royal Hospital for Children, Glasgow G51 4TF, UK; sarah.adam@ggc.scot.nhs.uk (S.A.); Barbara.Cochrane@ggc.scot.nhs.uk (B.C.); 3Royal Victoria Infirmary, Newcastle upon Tyne NE1 4LP, UK; sandraadams2@nhs.net; 4Sheffield Children’s NHS Foundation Trust, Sheffield S10 2TH, UK; heather.allen10@nhs.net (H.A.); lucy.white19@nhs.net (L.W.); 5Cardiff and Vale UHB, Cardiff CF14 4XW, UK; Sarah.Bailey3@wales.nhs.uk; 6Wrightington, Wigan and Leigh NHS Foundation Trust, Wigan WN1 2NN, UK; janette.banks@nhs.net; 7University College London Hospitals NHS Foundation Trust, London WC1N 3BG, UK; harriet.churchill@nhs.net (H.C.); c.ellerton@nhs.net (C.E.); 8Guy’s and St Thomas’ NHS Foundation Trust, London SE1 7EU, UK; Jennifer.Cook@gstt.nhs.uk (J.C.); Sarah.Firman@gstt.nhs.uk (S.F.); 9University Hospitals Birmingham NHS Foundation Trust, Birmingham B15 2TH, UK; Clare.Dale@uhb.nhs.uk (C.D.); Louise.robertson@uhb.nhs.uk (L.R.); 10Great Ormond Street Hospital for Children NHS Foundation Trust, London WC1N 3JH, UK; marjorie.dixon@gosh.nhs.uk (M.D.); Ide.Herlihy@gosh.nhs.uk (I.H.); Chloe.Millington@gosh.nhs.uk (C.M.); Rachel.skeath@gosh.nhs.uk (R.S.); 11Royal Hospital for Sick Children, Edinburgh EH9 1LF, UK; carolyn.dunlop@nhslothian.scot.nhs.uk; 12University Hospital Southampton NHS Foundation Trust, Southampton SO16 6YD, UK; Anita.Emm@uhs.nhs.uk; 13The National Society for Phenylketonuria, Purley CR8 9DD, UK; Suzanne.ford@nspku.org; 14University Hospitals of Leicester NHS Trust, Leicester LE1 5WW, UK; moira.french@uhl-tr.nhs.uk; 15Evelina London Children’s Hospital, London SE1 7EH, UK; Joanna.gribben@gstt.nhs.uk; 16Belfast Health & Social Care Trust, Belfast BT9 7AB, UK; Anne.Grimsley@belfasttrust.hscni.net (A.G.); joy.mcdonald@belfasttrust.hscni.net (J.M.); nicola.mcstravick@belfasttrust.hscni.net (N.M.); 17Sheffield Teaching Hospitals NHS Foundation Trust, Sheffield S5 7AU, UK; melanie.hill13@nhs.net (M.H.); catharine.noble@nhs.net (C.N.); 18Royal Liverpool and Broadgreen University Hospitals NHS Trust, Liverpool L7 8XP, UK; shirley.judd@rlbuht.nhs.uk; 19Ninewells Hospital, Dundee DD1 9SY, UK; karen.lang@nhs.net; 20Aneurin Bevan University Health Board Wales, Newport NP20 2UB, UK; Jo.males@wales.nhs.uk; 21Bristol Royal Hospital for Children, Bristol BS2 8BJ, UK; Camille.Newby@uhbw.nhs.uk (C.N.); Abigail.Robotham@uhbw.nhs.uk (A.R.); 22Norfolk and Norwich University Hospital, Norwich NR4 7UY, UK; rachel.pereira@nnuh.nhs.uk; 23Royal Aberdeen Children’s Hospital, Aberdeen AB25 2ZG, UK; Kathleenross@nhs.net; 24University Hospital of Wales, Cardiff CF14 4XW, UK; kath.singleton@wales.nhs.uk; 25Alder Hey Children’s NHS Foundation Trust, Liverpool L12 2AP, UK; allyson.terry@alderhey.nhs.uk; 26Royal Manchester Children’s Hospital, Manchester M13 9WL, UK; Karen.VanWyk@mft.nhs.uk (K.V.W.); fiona.white@mft.nhs.uk (F.W.); 27Bradford Teaching Hospitals NHS Foundation Trust, Bradford BD5 0NA, UK; joannewildgoose@hotmail.com; 28Salford Royal NHS Foundation Trust, Salford M6 8HD, UK; Alison.Woodall@srft.nhs.uk

**Keywords:** phenylketonuria (PKU), consensus, Delphi method, food labelling, phenylalanine, Phe, protein, exchanges

## Abstract

In phenylketonuria (PKU), variable dietary advice provided by health professionals and social media leads to uncertainty for patients/caregivers reliant on accurate, evidence based dietary information. Over four years, 112 consensus statements concerning the allocation of foods in a low phenylalanine diet for PKU were developed by the British Inherited Metabolic Disease Dietitians Group (BIMDG-DG) from 34 PKU treatment centres, utilising 10 rounds of Delphi consultation to gain a majority (≥75%) decision. A mean of 29 UK dietitians (range: 18–40) and 18 treatment centres (range: 13–23) contributed in each round. Statements encompassed all foods/food groups divided into four categories based on defined protein/phenylalanine content: (1) foods high in protein/phenylalanine (best avoided); (2) foods allowed without restriction including fruit/vegetables containing phenylalanine ≤75 mg/100 g and most foods containing protein ≤0.5 g/100 g; (3) foods that should be calculated/weighed as an exchange food if they contain protein exchange ingredients (categorized into foods with a protein content of: >0.1 g/100 g (milk/plant milks only), >0.5 g/100 g (bread/pasta/cereal/flours), >1 g/100 g (cook-in/table-top sauces/dressings), >1.5 g/100 g (soya sauces)); and (4) fruit/vegetables containing phenylalanine >75 mg/100 g allocated as part of the protein/phenylalanine exchange system. These statements have been endorsed and translated into practical dietary management advice by the medical advisory dietitians for the National Society for PKU (NSPKU).

## 1. Introduction

Phenylketonuria (PKU) is a rare, inherited metabolic disorder (IMD) caused by phenylalanine hydroxylase deficiency, leading to an abnormal accumulation of blood phenylalanine. Without treatment, it causes severe and irreversible intellectual disability. However, national newborn screening programmes detect PKU, which enables treatment to commence in early infancy with outcomes associated with a broad range of normal general ability. In the UK, the only available treatment is a rigorous, life-long dietary restriction of natural protein (i.e., meat, eggs, fish, cheese, nuts, bread, flour, pasta) in order to control blood phenylalanine levels [1,2] within the target range, as pharmacological treatments are not reimbursed by the National Health Service. Individuals with PKU tolerate only a limited amount of natural protein, with the amount individually determined. It is estimated that 80% of those with classical PKU are prescribed less than 10 g/day of protein [3].

Whilst the goal of dietetic management is essentially the same, the method for allocating phenylalanine/protein intake varies between and sometimes within countries, with insufficient evidence to recommend any one method [4,5,6,7,8,9]. Broadly there are two different methods, each with its merits and drawbacks: (1) patients are allocated a daily amount of phenylalanine/protein from all food, which is calculated to provide the prescribed phenylalanine intake; and (2) a phenylalanine/protein exchange system whereby amounts of food are calculated for a defined amount of phenylalanine (ranging from 10–50 mg of phenylalanine for each exchange) or protein (e.g., 1 g protein = 1 exchange). One food exchange can be replaced with an alternative exchange of an equivalent phenylalanine/protein amount.

In the UK, natural protein is apportioned using a protein exchange system whereby one exchange is equivalent to the amount of food that is calculated/measured to provide 1 g protein or 50 mg of phenylalanine. Individual tolerance based on blood phenylalanine levels determines the number of exchanges allocated, but is typically between 3–10 g exchanges/day (or 150–500 mg/day phenylalanine). The diet is then supplemented with a phenylalanine-free/low-phenylalanine protein substitute, foods naturally low in phenylalanine/protein, and special low protein foods (SLPF) (e.g., bread, flour, pasta) approved by the Advisory Board for Borderline Substances (ACBS), and prescribed by the General Practitioner (GP). Overall dietary management is complex, and a high degree of patient/caregiver knowledge and application is required to effectively implement dietary care. Successful dietary treatment is hampered by inconsistent dietary advice associated with unclear and historical recommendations, lack of comprehensive food phenylalanine analysis, similar plant species containing a variable phenylalanine content per 100 g of food, and an overwhelming range of manufactured foods with unclear declarations of protein content. In the UK, national practical guidance about the implementation of dietary treatment in PKU has not been reviewed since 1993 [10].

Patients with PKU are cared for by multidisciplinary teams of health professionals working across many care settings within the UK. Almost every professional involved has an opinion about the practical implementation of dietary management. This causes wide variability in dietary advice given by practitioners, leading to uncertainty for patients and caregivers and inexperienced health professionals seeking advice from peers. In addition, patients/caregivers readily turn to social media for information that may be based on the erroneous interpretations of others. Multiple sources of discordant information are also available through apps and specialist manufacturers of dietary products. International dietary practices also vary, adding another tier of misunderstanding and ambiguity [4]. It is important that patients with PKU and their caregivers receive reliable and uniform dietary information from their health care professionals.

In 2016, the British Inherited Metabolic Disease Dietitians Group (BIMDG-DG) identified several controversial issues concerning the calculation of protein in a low phenylalanine diet [11]. This included EU legislation that foods containing a protein content of ≤0.5 g/100 g do not need to declare their specific protein content on food labels [12]. This has caused considerable confusion about which foods could be permitted within a phenylalanine restricted diet. There was also discrepancy about the allocation of fruits and vegetables according to their phenylalanine/protein content. This led to the development of 23 BIMDG-DG national consensus statements about the interpretation of protein food labelling and allocation of foods in a low phenylalanine diet [11].

Since the publication of the first report [11], the BIMDG-DG have continued to develop new statements about food allocation in a low phenylalanine diet utilising Delphi methodology. This time, all food categories have been systematically examined, and a comprehensive set of consensus guidelines for interpreting their protein content and allocating these foods within a low phenylalanine diet has been developed.

## 2. Materials and Methods

Over a 4-year period (November 2015 to September 2019), 112 consensus statements from UK dietitians working in PKU were developed regarding the inclusion, exclusion, and allocation of foods within a low phenylalanine diet *specifically for use by dietitians.* These were aimed at restrictions of protein of ≤10 g/day or phenylalanine ≤500 mg/day. These statements considered most commercially available foods as well as special low protein foods. The process included 10 rounds of Delphi consultation to gain a majority decision in a structured and systematic way [13]. It involved an independent facilitator collecting the opinion of clinical dietitians working with PKU. The facilitator issued proposed statements about food allocation, each time presenting any available research evidence along with the protein/phenylalanine content of foods. Questionnaire responses in each round were gathered and results discussed by telephone conference. Metabolic dietitians from the BIMDG voted on each statement. The process was repeated, with modification of the statements, until there was at least 75% agreement (an arbitrary figure chosen to represent a majority decision). Any unresolved statements at round one of the Delphi process were carried over to the next round of statements for further discussion. A more detailed description of this process has been reported previously [11].

All foods/food groups were systematically divided into nineteen subgroups with a mean of six statements per subgroup (range: 1–14): milk and milk replacements (*n* = 2); dairy products and alternatives (*n* = 12); breads and cereals (*n* = 4); spreads and dips (*n* = 5); sauces and soups (*n* = 13); pasta and rice (*n* = 2); potato and potato products (*n* = 6); fruit and vegetables (n = 7); meat and alternatives (*n* = 7); drinks (*n* = 8); sweet snacks (*n* = 13); savoury snacks (*n* = 10); sugars, sweeteners and syrups (*n* = 9); herbs and spices (*n* = 1); low protein special foods (*n* = 2); flours and starch (*n* = 5); baking ingredients (*n* = 4); gluten-free products (*n* = 1); and gelatine containing products (*n* = 1).

No ethical approval was required for this project as it is not considered research as defined by the UK Policy Framework for Health and Social Care Research [14]. Descriptive analysis was used to present the results.

## 3. Results

In the UK, there are nine paediatric and nine adult PKU specialist centres for PKU, with around 18 district general hospitals who share PKU care with the specialist centres.

In total, 93 BIMDG dietitians (57% (*n* = 53) paediatric; 32% (*n* = 30) adult; 11% (*n* = 10) working with both adults and children) from all represented UK PKU treatment centres (*n* = 34) responded to the consensus statements during the four year study period. In each round of the Delphi process, there were approximately 70 active BIMDG dietetic members potentially able to participate. The numbers of dietetic contributors varied in each round due to the movement of dietitians (maternity leave, retirement, illness, secondment, or specialty change). A median of 29 dietitians (range: 18–40) and 18 centres (range: 13–23) contributed to each round. All specialist UK PKU IMD centres were represented in each round except for one adult centre only able to participate in two of 10 rounds. All BIMDG dietitians received copies of the results for each round and the minutes of meetings. Fifty-eight percent (*n* = 65/112) of statements received 100% agreement, a further 26% (*n* = 29/112) received ≥90% agreement, and 15% (17/112) were between 79–89%.

Of the 112 statements, most had majority agreement within one round of the Delphi process. Only nine statements received less than 75% agreement on the first round of discussion (fruit/vegetables, low protein milks, coconut desserts, soya sauce, special low protein foods, vegetable crisps, eggs, cheese, and seeds). These statements were then modified and reconsidered in the next round. New statements for other foods continued to be added at each round. Consensus was agreed in seven of nine statements on the second round of discussion. Two statements (fruit/vegetables and low protein milks) required discussion, modification, and voting over three consecutive rounds to achieve consensus. No statements required deletion, only modification, as all foods/food groups were required to have a statement.

Each food subgroup was systematically discussed via the Delphi process and allocated into one of four categories (with further subcategories), based on their protein or phenylalanine content.

Category 1: Foods high in protein or phenylalanine that are best avoided (Table 1). This included two subcategories: 

(a) Foods high in protein (generally containing protein >15 g/100 g).

(b) Foods containing aspartame and therefore phenylalanine.

Category 2: Foods allocated without restriction or measurement (defined as exchange-free foods) (Table 2). Generally, any foods with an upper protein content cut-off point ≤0.5 g/100 g or containing exchange-free ingredients were considered exchange-free. Exceptions included spices (with a higher protein content) that are used for flavouring purposes only and consumed in small quantities. Overall, this category consisted of four subcategories:

(a) Fruits and vegetables containing phenylalanine ≤75 mg/100 g.

(b) Manufactured foods containing protein ≤0.5 g/100 g or exchange-free ingredients.

(c) Manufactured food containing protein >0.5 g/100 g, but used in small amounts so provide minimal contribution to protein intake.

(d) SLPF containing exchange-free ingredients or a phenylalanine content <25 mg/100 g that have been ACBS approved and are available on prescription for low protein diets.

Category 3: Manufactured foods/SLPF allocated as part of the protein/phenylalanine exchange system according to their protein/phenylalanine content per 100 g (Table 3). One protein/phenylalanine exchange is the amount of food that is calculated/measured to provide either 1 g of protein or 50 mg of phenylalanine from its food analysis. This category included any SLPF and manufactured foods with a phenylalanine/protein content above the upper cut-off point (>0.5 g/100 g) and containing protein exchange ingredients (e.g., milk, wheat, flour, rice, egg, and soya). This section was divided into four subcategories with different protein/phenylalanine upper cut-off points based on the food portion size that would typically be consumed or if they contained a high proportion of exchange-free fruit or vegetables (e.g., commercial cooking sauces): 

(a) Liquid plant milks and animal milks that contain protein >0.1 g/100 mL or specialist low protein milks with a phenylalanine content >5 mg/100 mL that have been approved by the UK ACBS and available on prescription for low protein diets. The upper protein/phenylalanine cut-off is set at a low amount due to potential high daily volumes that may be consumed.

(b) Manufactured foods containing protein >0.5 g/100 g or SLPF containing phenylalanine >25 mg/100 g and containing exchange ingredients. This group contained most of the manufactured foods. It also included some SLPF approved by the UK ACBS and available on prescription for low protein diets.

(c) Commercial sauces and tabletop sauces containing protein >1 g/100 g and exchange ingredients. This subcategory mainly consisted of commercial sauces containing vegetables, which have a lower phenylalanine content per 1 g of protein than cereals or animal foods [15]. This group also included cake decorations, as the amounts consumed are small.

(d) Soya sauces containing protein >1.5 g/100 g. This subcategory allowed soya sauce with a higher amount of protein because the amount consumed is small and only a few brands contained protein less than this amount.

Category 4: Fruit/vegetables containing phenylalanine >75 mg/100 g allocated as part of the protein/phenylalanine exchange system (Table 4). This included vegetable crisps prepared from exchange-free vegetables that had a higher phenylalanine content and were discussed in Evans et al. [11]. Potatoes have a lower phenylalanine content than 75 mg/100 g but are calculated/measured as exchange foods due to the amount consumed daily in the UK diet.

All statements were accepted and endorsed by the NSPKU. The statements were then translated into easier guidance for patients and carers [16]. A detailed list of all statements is provided in Appendix A.

## 4. Discussion

This paper reports the results of four years of in-depth national discussions amongst experienced UK metabolic dietitians using Delphi methodology to gain consensus on the suitability and allocation of foods in a low phenylalanine diet, for patients tolerating a natural protein intake of ≤10 g/day. Having uniform national recommendations across all UK centres treating PKU should enable health professionals and support groups to provide consistent information to patients with PKU. Securing consensus amongst health professionals was challenging, but essential, as there were many differing opinions leading to disparate patient information and unfounded dietary practices. The Delphi methodology was systematic, impartial, and consistent, involving representation from all major UK PKU centres. All terms of reference were agreed in advance and an impartial facilitator ensured the process was conducted transparently and without bias [13]. Eighty-four percent (94/112) of statements received at least 90% agreement.

Historically in the UK, exceptionally low protein foods such as sugar, jam, honey, and vegetable oils have been permitted as exchange-free and there is no evidence to suggest this advice should change. There was much discussion about the allocation of protein cut off points for different food groups and categorisation of foods/drinks within each group considering the impact on blood phenylalanine control if foods were not calculated/measured as part of the phenylalanine exchange system. It was clear that not all food subgroups could be considered in the same way, dependent on the weight of food that would be consumed and the role of each food within the diet. Most manufactured foods were defined as exchange-free only if they had a protein content ≤0.5 g/100 g of food. However, even within this subcategory, some foods required additional rules to prevent over consumption when protein content was close to the upper cut off point, and the amount consumed would exceed 100 g in one portion. This applied to dairy-free yoghurts and desserts based on plant milks that contained a protein content of around 0.5 g/100 g.

Since the 1960s, UK health professionals have given most fruits and vegetables (except potatoes) with a phenylalanine content ≤75 mg/100 g as exchange-free in a low phenylalanine diet. This guidance was reconsidered when developing the consensus statements, as evidence from a series of studies indicated that this maximum cut off did not adversely impact blood phenylalanine control in PKU [17,18,19,20]. This was also consistent with the 2017 PKU European Guidelines [21] and in turn, this influenced other consensus statements. For example, many cook-in/pour over sauces are primarily made from exchange-free vegetables and other exchange-free ingredients such as starches and seasonings only. It was therefore decided to calculate/measure vegetable sauces as part of the phenylalanine exchange system only if the sauce contained exchange ingredients (such as cream, flour) and a protein content >1 g/100 g. Soya sauce was considered exchange-free if the protein content was ≤1.5 g/100 g. A higher cut off point was given for soya sauce, as the amount used in recipes is generally small.

Plant milks and special milk replacements were a group that also required more specific definition of protein/phenylalanine cut off points due to potentially high daily volumes being consumed, thereby contributing a significant amount of protein/phenylalanine. A stringent exchange-free upper limit of ≤0.1 g protein/100 mL was set for regular and plant milks, and an upper limit of phenylalanine of ≤5 mg/100mL for low protein special milks available on ACBS prescription. This was because the volumes consumed may be high when taking this as a regular hot or cold drink, ‘milk’ shake or ‘latte’. Only a small number of plant milks (some coconut milks) and low protein special milks (protein-free only) can therefore be given without calculation/measurement in the diet.

Currently, around 10% of SLPF contain more phenylalanine than the upper cut off point (≤25 mg/100 g) due to added ingredients such as milk, seeds, and rice flour [22]. It is hoped that these consensus statements will encourage manufacturers of SLPF to develop new low protein special foods with a phenylalanine content of ≤25 mg/100 g, so that they can be eaten without measurement or restriction. There seems to be little value in having SLPF that must still be limited and controlled within the diet. All SLPF should have clear labelling identifying their phenylalanine content (per 100 g of raw ingredients and per 100 g after preparation) with the full list of ingredients given, and any protein containing ingredients identified in bold on the ingredients list [23].

There were study strengths and limitations. Although dietitians represented both paediatric and adult care, they had differing opinions on the stringency of dietary guidance needed. Whilst this added to discussions, it also increased the challenge of producing statements that would meet the needs of the majority of the PKU population. Although all BIMDG dietitians had the opportunity to comment on each statement and vote in each round of the Delphi process, due to career changes or other circumstances such as maternity leave, the dietitians responding in each round were not necessarily the same each time. Even so, a representative from almost all major IMD specialty treatment centres (in both paediatric and adult care) was represented in each round. Overall, 100% consensus was reached for 64 statements. It was considered impractical to aim for 100% agreement for all statements, but a consensus cut off of 75% was agreed upon as it represented the majority of opinion. It may have been valuable to seek patient/carer opinion on each statement. Pragmatically, it was considered important as a first step to gain dietetic consensus as this was considered a substantial barrier to consistency of care, before translating into practical guidance for patients.

It is acknowledged that internationally, different systems are used to calculate/measure protein/phenylalanine in the PKU diet, each with its own inherent weaknesses [4,5,6,7,8,9]. Using upper protein/phenylalanine cut-off points has the disadvantage of having to measure/calculate foods as part of an exchange system if they contain protein marginally over the cut-off, whilst eating foods as exchange-free if they are just under the cut-off point, but it does give direct guidance and allows many foods to be eaten without measurement or calculation. Overall, these statements are designed to provide broad guidance for dietitians with limited experience in treating PKU on how to safely manage individuals with PKU. It is also important that they understand the foundation for the statements. The proposed guidance should be used in conjunction with individual tailored advice considering patient food likes, aversions, and level of understanding.

The statements were designed to be used by dietitians and heath care professionals. However, since completion of the statements, the NSPKU medical advisory dietitians have developed a dietary information guide using the statements as a basis for practical advice [16]. The next step will be to evaluate the adherence to these statements by both health professionals and patients with PKU and their caregivers.

## 5. Conclusions

The development and publication of UK consensus statements for food labelling and protein allocation in the PKU diet is an important step in harmonising dietary advice and effecting consistency of care for patients with PKU. Developing these statements in partnership with BIMDG dietitians using Delphi methodology has ensured that all dietitians have had the opportunity to participate in the development in an impartial, transparent, and consistent process. In the UK, this is the first time that such extensive agreement has been reached amongst specialist dietitians, with the results being simplified and implemented into patient care. It is also hoped that these guidelines will be respected and adopted by manufacturers of SLPF to ensure patients with PKU can gain full advantage from consuming low protein foods in a low phenylalanine diet.

## Figures and Tables

**Table 1 nutrients-12-02205-t001:** Category 1: Foods high in protein or phenylalanine.

1. Foods High in Protein or Phenylalanine that Are Best Avoided	
(a) Foods high in protein (approximately >15 g/100 g)	Meat, fish, eggs, nuts, cheeses, seeds, soya products, Quorn, goji berries, peanut butter, tofu, spreadable yeast extracts. Exceptions: soft cheeses, soya cheese, baked products containing seeds or eggs as an ingredient, baked goods with eggs as an ingredient - these fall into category 3b and are used as part of the exchange system. Eggs contain protein <15 g/100 g. Although they are used as part of the protein exchange system in baked goods, one hen’s egg is high in protein and best avoided.
(b) Foods containing aspartame	Aspartame containing food and drinks (e.g., fizzy drinks, fruit juice, fruit tea, milkshake powders/syrup, smoothies, squash, chewing/bubble gum, desserts, jelly, sweets, tabletop sweeteners).

**Table 2 nutrients-12-02205-t002:** Category 2: Foods allocated without restriction.

2. Foods Allocated without Restriction or Measurement (Defined as Exchange-Free Foods)	
(a) Fruits and Vegetables containing phenylalanine ≤75 mg/100 g (except potatoes and vegetable crisps)	Apples, apricots, avocado, bananas, banana chips, bilberries, blackberries, blueberries, candied angelica, candied peel, cherries, clementines, cranberries, currants, custard apples, damsons, dates, dragon fruit, fruit crisps (e.g., apple, pineapple), fruit pie filling, fruit mincemeat, fruit salad, glacé cherries, gooseberries, grapes, grapefruit, greengages, guavas, jackfruit, kiwi fruit, kumquats, lemons, limes, loganberries, lychees, mandarins, mango, medlars, melon, nectarines, olives, oranges, papaya (paw paw), peaches, pears, physalis, pineapple, plums, pomegranate, prickly pear, prunes, quince, raisins, raspberries, rhubarb, satsumas, Sharon fruit, star fruit, strawberries, sultanas, tamarillo, tangerines, watermelon. Artichoke, aubergine, baby corn, beetroot, cabbage, capers, caperberries, carrots, cassava, celeriac, celery, chayote, chicory, courgette, cucumber, dudhi, eddoes, endive, fennel, garlic, gherkin, ginger, green beans (dwarf, French, runner), karela, kohlrabi, leeks, lettuce, marrow, mooli, mushrooms, okra, onion, pak choi, parsnips, peppers, pickled vegetables (e.g., onion, gherkins, red cabbage), plantain, pumpkin, radish, salad cress, samphire, squash (butternut, acorn, spaghetti), swede, sweet potato, tomato, turnip, watercress, water chestnuts. Fruit and vegetable-based foods containing exchange-free fruits/vegetables and other exchange-free ingredients (e.g., frozen or canned fruit/vegetables, tomato puree/passata).
(b) Manufactured foods containing protein ≤0.5 g/100 g or exchange-free ingredients	Sugar (brown, cane, caster, demerara, fruit, glucose, granulated, icing, molasses, muscovado, white). Jam, honey, marmalade, syrup (agave, fruit, golden, maple, treacle). Fats (oils, oil sprays, ghee, lard). Baking ingredients (arrowroot, baking powder, bicarbonate of soda, cassava/tapioca flour, cornflour/maize starch, cream of tartar, sago). Aspartame-free drinks (squash, fruit drinks, soft drinks, black/herbal tea and coffee). Artificial sweeteners (except aspartame). Condiments (mint jelly, mint sauce, salt, vinegar). Fibres/gums (e.g., psyllium fibre/husks, xanthan gum). Aspartame-free milkshake powders/syrups and custard powder containing exchange-free ingredients. Plants & cereals (konnyaku, sago, tapioca, cassava crisps).
(c) Manufactured food containing protein >0.5 g/100 g but used in small amounts	Fats (butter, margarine). Herbs, spices, condiments (e.g., pepper). Food colouring and flavourings/essences.
(d) Special low protein foods containing exchange-free ingredients or a phenylalanine content <25 mg/100 g	Low protein: bread (sliced, rolls, baguettes), biscuits, breakfast cereals, cereal bars, cakes, chocolate, chocolate spread, cheese sauce, crackers, cake mix, dessert/custard mixes, egg replacer, fish substitute mixes, flour, pizza bases, pasta, rice, sausage/burger mixes. Low protein milk replacement Prozero (Vitaflo). Includes most UK ACBS prescribed low protein products.

**Table 3 nutrients-12-02205-t003:** Category 3: Manufactured foods allocated as part of an exchange system based on protein or phenylalanine content.

3. Manufactured Foods Allocated as Part of the Protein Exchange System According to Their Protein/Phenylalanine Content per 100 g	
(a) Liquid plant and animal milks with protein >0.1 g/100 g or 0.1 g/100 mL or specialist low protein milks with phenylalanine >5 mg/100 mL	Animal milks (e.g., cow, goat, sheep), full fat, semi-skimmed, skimmed, condensed. Plant milks (e.g., coconut, oat, almond, soya). Includes coffee with these added (e.g., lattes, cappuccino, frappuccino, macchiato, coffee pods/sachets). Low protein milk replacements–Dalia 6.4 mg Phe/100 mL (Taranis), Lattis 12 mg Phe/100 mL (Mevalia), Loprofin 10 mg Phe/100 mL and SnoPro 8.7 mg Phe/100 mL (Nutricia).
(b) Foods containing protein >0.5 g/100 g or specialist low protein foods containing phenylalanine >25 mg/100 g and containing exchange ingredients	Bread and bread products, biscuits and cakes made from regular flour, butter, cheese spread, cream, cream cheese, chocolate spreads, cocoa powder, breakfast cereals, cereal grains, cereal bars, cereal products (pancakes, waffles, stuffing, Yorkshire pudding), chocolate, coconut based desserts and products, corn/rice based snacks, crackers, cream, dairy desserts (custard, instant, fromage frais, mousse), dips (sweet & savoury), drinking chocolate, flour and flour products, free-from and vegan/plant cheeses, fondant icing, fruit bars, fudge, gelatine containing foods, gluten-free foods, gravy, herb/spice rubs and coatings, hummus, ice cream (dairy/non-dairy), ice lollies, icing/frosting, jelly, legumes/pulses (baked beans, lentils), lemon curd, liquorice, marshmallows/mallows, marzipan, milk based sauces, milkshake powders/syrups, mustard, nut spread, pasta/noodles, pesto, plant/vegetable spreads, popcorn, potato crisps, pretzels, pot noodles, puddings/desserts, rice, rice/oat cakes, soft cheese, sorbets, soups, stock cubes, sweets, tapenade, toffee, tofu, vegan meat/fish or egg alternatives, vegetable crisps, vegetable soups, yoghurt (dairy/non-dairy). Note: yogurts, dairy desserts and coconut-based puddings with a protein content ≤0.5 g/100 g should be limited to 1 per day. Low protein ACBS prescription products: Promin potato pots, Promin potato cakes, Taranis fish substitute.
(c) Commercial sauces and tabletop sauces containing protein >1 g/100 g and containing exchange ingredients	Cook-in, pour-over or liquid sauces (curry, sweet & sour, tomato, vegetable), oil-based dressings (mayonnaise, salad cream, vinaigrette), table top sauces (brown, chilli, chutney, horseradish, mint, pickles, tartare, tomato ketchup). Cake decorations/sprinkles.
(d) Soya sauces containing protein >1.5 g/100 g	Most soya sauces have protein >1.5 g/100 g.

**Table 4 nutrients-12-02205-t004:** Category 4: Exchange fruit and vegetables (phenylalanine >75 mg/100 g).

4. Exchange Fruit/Vegetables Containing Phenylalanine >75 mg/100 g Allocated as Part of the Protein Exchange System According to Their Phenylalanine/Protein Content per 100 g	
Fruit & Vegetables with phenylalanine content 75–99 mg/100 g	Figs Asparagus, bamboo shoots, beansprouts, broccoli, brussels sprouts, cauliflower, mange tout, sugar snap peas, whole hearts of palm. A standard portion size of 60 g is used for 1 phenylalanine exchange.
Fruit & Vegetables with phenylalanine content >100 mg/100 g	Passionfruit Broad beans, chestnuts, choi sum, corn on the cob, kale, mixed vegetables, peas and petit pois, romanesco, rocket, spinach, spring greens, sweetcorn kernels, sweet potato fries with coating, vine leaves, yams. Phenylalanine content is used to determine amount for 1 phenylalanine exchange.
Potatoes	All potatoes and potato products.
Vegetable crisps	All vegetable crisps (except cassava).

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
