# Peer review of "Uniformity of Food Protein Interpretation Amongst Dietitians for Patients with Phenylketonuria (PKU): 2020 UK National Consensus Statements"

_nutrients, 2020, doi:10.3390/nu12082205_

Round 1
Reviewer 1 Report
General comments
It is desirable to have a standardized approach, categorization and naming system to improve consistency for treatment advice. In this well-written manuscript, the authors describe a comprehensive approach to this end for dietary advice in the management of PKU in the UK. This manuscript would seem to be a useful reference for that group to use in order improve the consistency in treating PKU in the UK. The process used to achieve the consensus statements seems very inclusive and transparent. I have a few comments and questions detailed below (see Specific Comments) that look to clarify some points.
The authors are likely aware that different counting systems are used outside of the UK. Just a few more additional background details about the system used in the UK would make this publication more understandable to "outsiders" without having to look at secondary references or infer from the appendix. Several of my comments below speak to this frame of reference and as such could be taken with a grain of salt! Further, the authors admit that "Overall dietary management is complex" (line 82), without mentioning that other systems in widespread use might be far less complex, nor did the Delphi method look at the overall complexity of the UK system.
Specific comments
Introduction
- (Line 81) "Natural protein is apportioned using a protein ‘exchange’ system whereby 1g protein is equivalent to one exchange": This is not a clear description, especially when some readers might be more familiar with the American system using 1 exchange = 15 mg phenylalanine. It is not until line 162 where the system is more clearly explained that "1 exchange = the amount of food that is calculated/measured to provide 1g protein". I suggest the description in line 81 be clarified, or perhaps even expanded into a description of the exchange system that is complete enough to obviate the need to refer to other publications or the appendix.
Materials and methods
- Delphi consultation is an appropriate method to achieve consensus within a group. However, the results obtained are limited by the members of the Delphi process. If one of the goals of the process in this case is to increase certainty of patients/caregivers it would be important to include that group in the Delphi process. As stated in the introduction (line 83), "a high degree of patient/caregiver knowledge and application is required to effectively implement dietary care". Without seeking input from patients/caregivers you are only guessing if the resulting system is likely to result in an improvement. This is a major limitation of the study design.
- (line 118) "the process included 10 rounds of Delphi consultation": From the results section (line 151) it appears that there were 3 rounds.
- (line 124): it is not clear if there was a mechanism to remove any statements from the list
Results
- (line 139) "34 UK PKU treatment centres": It would be useful to know what proportion of all treatment centres in the UK this represents.
- (line 143) "All but one designated adult UK PKU IMD centre was represented in each round": It would be useful to know how many adult UK PKU centres there are in total, and to expand on this statement to clarify in what way was this particular centre not represented (did it participate at all?).
- (line 138 vs 142): The first statement mentions 93 BIMDG dietitians and then the second statement mentions 70 "current" dietetic BIMDG members. The sentence (Line 142) "A median of 29 dietitians (range:18-40) and 18 centres (range:13-23) from approximately 70 current dietetic BIMDG members contributed to each round" now becomes unclear, as it implies the exclusion of some of the 93 respondents. If the authors feel that it is important to mention only 70 of the original 93 respondents are still "current", it should be done in a separate statement.
- The next several comments mostly reflect that understanding the UK system in the first place is key to making sense of the results of the Delphi process. While it is reasonable to expect a certain amount of background knowledge of the reader (or their willingness to look at secondary references for clarification), I found that this manuscript failed to frame the results in such a way that an average "expert" in PKU from outside the UK system could make sense of the categories. This is in contrast to the 2019 paper (ref #6) by the same group that provided much more explanation and rationale for the categories within the text and tables of the manuscript.
-
- (line 156 – category 1 foods) "Foods high in protein": This is vague and inaccurate. If possible, it is better to state a specific cut-off (e.g., >"x"g/100g), similar to those stated in category 3. This group also includes low protein – high phenylalanine (aspartame containing) products, hence my "inaccurate" comment.
- (line 157 – category 2 foods) Given the description of category 3 subgroup 1 (foods >0.1g protein/100g), this category (category 2, "foods very low in protein/phenylalanine") would at first blush appear to be foods with ≤ 0.1g protein/100 g or mL. It is only upon examining Table 1 that the reader might be surprised to find that the cut-off is more like 1.5g protein/100g for some foods in this group.
- (line 164 – category 3 foods) At first glance, it is not clear what advantage there is in sub-dividing this category. According to the definition "Foods that should be calculated/weighed as an exchange food", the foods all need to be calculated into the exchange system, so the sub-classification does not seem to provide a meaningful distinction. This comment was made thinking that the sub-group labels were meant to be protein ranges, i.e. >0.1 to 0.5, >0.5 to 1, >1 to 1.5 and >1.5 to ??. After finishing the manuscript, the appendix, other references, and looking the NSPKU website (for Ref 10), the sub-grouping made more sense.
- (line 176 – category 4 foods) It is not clear why "foods with a known phenylalanine content" would be separately accounted for in this system. All foods have a "knowable" phenylalanine content given sufficient effort. The phenylalanine level cut-offs for subgroups in this category roughly align with the protein level cut-offs for category 3 levels (25 mg PHE with 0.5g protein, 5 mg PHE with 0.1g protein, 75 mg PHE with 1.5g protein), so it appears very arbitrary to use separate categories. It seems like this group would more accurately be categorized as foods specifically manufactured or marketed to be low-protein… plus some random vegetables (why is this sub-group not worked into category 3?).
-
- The descriptions (line 164, line 180-181 and the matching table 1 description for category 4) that state "based on… and if" are not clear (e.g., in the sentence "I categorize my sweaters based on colour, weight, and if they contain wool", the "and if" is not a Boolean operator for the list). Conversely, the Table 1 description for category 3 ("based on…but only if") is clearly worded.
Discussion
- (lines 242-247): While the focus of this manuscript obviously is not the strengths and weaknesses of the various methods to administer a low phenylalanine diet, there should be some acknowledgement that certain issues are peculiar to certain counting systems. The weakness of any system that sets arbitrary thresholds for counting or not counting certain foods is highlighted by this section. For example, foods with phenylalanine of 23 mg /100g compared to 27 mg /100g are not all that different, but the strict (and arbitrary) cut-point of 25 mg/100g causes them to be treated differently. The European guidelines suggest the equally arbitrary cut-off of 50 mg/100g for low-protein foods. Making the statement "There seems little value in having low protein foods that must still be limited and controlled within the diet" is purely an artifact of this counting system, especially if you consider the "regular" version of that food might have 10 times the phenylalanine!
Author Response
Reviewer 1
General comments
It is desirable to have a standardized approach, categorization and naming system to improve consistency for treatment advice. In this well-written manuscript, the authors describe a comprehensive approach to this end for dietary advice in the management of PKU in the UK. This manuscript would seem to be a useful reference for that group to use in order improve the consistency in treating PKU in the UK. The process used to achieve the consensus statements seems very inclusive and transparent. I have a few comments and questions detailed below (see Specific Comments) that look to clarify some points.
Thank you for your positive comments.
The authors are likely aware that different counting systems are used outside of the UK. Just a few more additional background details about the system used in the UK would make this publication more understandable to "outsiders" without having to look at secondary references or infer from the appendix. Several of my comments below speak to this frame of reference and as such could be taken with a grain of salt! Further, the authors admit that "Overall dietary management is complex" (line 82), without mentioning that other systems in widespread use might be far less complex, nor did the Delphi method look at the overall complexity of the UK system.
Thank you for your feedback, we have added more detail in the introduction (paragraphs 2 and 3).
Specific comments
Introduction
- (Line 81) "Natural protein is apportioned using a protein ‘exchange’ system whereby 1g protein is equivalent to one exchange": This is not a clear description, especially when some readers might be more familiar with the American system using 1 exchange = 15 mg phenylalanine. It is not until line 162 where the system is more clearly explained that "1 exchange = the amount of foodthat is calculated/measured to provide 1g protein". I suggest the description in line 81 be clarified, or perhaps even expanded into a description of the exchange system that is complete enough to obviate the need to refer to other publications or the appendix.
We have added more detail in the early part of the introduction to explain the UK exchange system.
Materials and methods
- Delphi consultation is an appropriate method to achieve consensus within a group. However, the results obtained are limited by the members of the Delphi process. If one of the goals of the process in this case is to increase certainty of patients/caregivers it would be important to include that group in the Delphi process. As stated in the introduction (line 83), "a high degree of patient/caregiver knowledge and application is required to effectively implement dietary care". Without seeking input from patients/caregivers you are only guessing if the resulting system is likely to result in an improvement. This is a major limitation of the study design.
These guidelines were developed for dietetic use, not patient use. This has now been highlighted in the materials and methods in italics. The goal was to bring about dietetic consensus as this was a significant barrier for patients/carers. It was considered necessary for dietitians to agree first before involving patients/carers. The guidelines have since been translated and simplified into practical guidelines for patients and will be evaluated in the near future. However, we acknowledge that it might have been useful to have some feedback on patient/carer thoughts on the guidelines during the process and this has been added to the discussion limitations.
- (line 118) "the process included 10 rounds of Delphi consultation": From the results section (line 151) it appears that there were 3 rounds.
This has been clarified in the methods and results section. There were 10 rounds, but some foods were carried over to the next round as they required further discussion and voting. Two foods required discussion over 3 consecutive rounds to obtain consensus.
- (line 124): it is not clear if there was a mechanism to remove any statements from the list
No statements were removed, only modified as all foods/food groups were required to have a statement. This has been clarified in the text in the results section.
Results
- (line 139) "34 UK PKU treatment centres": It would be useful to know what proportion of all treatment centres in the UK this represents.
This represents almost all treatment centres in the UK. There are 9 paediatric and 9 adult specialist centres for UK PKU but around 18 district general hospitals who share care with the specialist PKU centres. There may be a small number of other hospitals who care for less than 10 PKU patients, but almost all patients with PKU in the UK are under the ‘umbrella’ care of a specialist centre. This has been clarified at the start of the results.
- (line 143) "All but one designated adult UK PKU IMD centre was represented in each round": It would be useful to know how many adult UK PKU centres there are in total, and to expand on this statement to clarify in what way was this particular centre not represented (did it participate at all?).
There are 9 adult specialist centres, with some additional district general hospitals who care for over 40 adults with PKU. The centre in question did respond in 2/10 rounds but the dietitians at this hospital have a high workload with a small number of metabolic dietitians. They had the opportunity to respond and received minutes of meetings. This has been clarified in the text in the results section.
(line 138 vs 142): The first statement mentions 93 BIMDG dietitians and then the second statement mentions 70 "current" dietetic BIMDG members. The sentence (Line 142) "A median of 29 dietitians (range:18-40) and 18 centres (range:13-23) from approximately 70 current dietetic BIMDG members contributed to each round" now becomes unclear, as it implies the exclusion of some of the 93 respondents. If the authors feel that it is important to mention only 70 of the original 93 respondents are still "current", it should be done in a separate statement.
There is constant change of dietitians in and out of the BIMDG dietitians’ group due to career changes, retirement, sickness and maternity leave. So in total 93 dietitians responded in at least one round of the process over the 4 year period. However, at any point in time there were around 70 dietitians in the BIMDG. This total number changed from round to round but was consistently around 70. BIMDG-dietitian group membership lists are not updated regularly to quantify this specifically for every round. We have qualified this in the text in the results section.
- The next several comments mostly reflect that understanding the UK system in the first place is key to making sense of the results of the Delphi process. While it is reasonable to expect a certain amount of background knowledge of the reader (or their willingness to look at secondary references for clarification), I found that this manuscript failed to frame the results in such a way that an average "expert" in PKU from outside the UK system could make sense of the categories. This is in contrast to the 2019 paper (ref #6) by the same group that provided much more explanation and rationale for the categories within the text and tables of the manuscript.
Thank you for you feedback. It is important to us that this paper is comprehensible for people outside of the UK so we appreciate your input. We have modified the text and included additional tables in the results.
- (line 156 – category 1 foods) "Foods high in protein": This is vague and inaccurate. If possible, it is better to state a specific cut-off (e.g., >"x"g/100g), similar to those stated in category 3. This group also includes low protein – high phenylalanine (aspartame containing) products, hence my "inaccurate" comment.
We have clarified that this includes foods high in protein or phenylalanine. We have added a cut-off point for protein to help define high protein foods.
- (line 157 – category 2 foods) Given the description of category 3 subgroup 1 (foods >0.1g protein/100g), this category (category 2, "foods very low in protein/phenylalanine") would at first blush appear to be foods with ≤ 0.1g protein/100 g or mL. It is only upon examining Table 1 that the reader might be surprised to find that the cut-off is more like 1.5g protein/100g for some foods in this group.
This section (Category 2) has been modified, further explained and a table (Table 2) added for more information.
- (line 164 – category 3 foods) At first glance, it is not clear what advantage there is in sub-dividing this category. According to the definition "Foods that should be calculated/weighed as an exchange food", the foods all need to be calculated into the exchange system, so the sub-classification does not seem to provide a meaningful distinction. This comment was made thinking that the sub-group labels were meant to be protein ranges, i.e. >0.1 to 0.5, >0.5 to 1, >1 to 1.5 and >1.5 to ??. After finishing the manuscript, the appendix, other references, and looking the NSPKU website (for Ref 10), the sub-grouping made more sense.
This section (Category 3) has been modified, further explained and a table (Table 3) added for more information.
- (line 176 – category 4 foods) It is not clear why "foods with a known phenylalanine content" would be separately accounted for in this system. All foods have a "knowable" phenylalanine content given sufficient effort. The phenylalanine level cut-offs for subgroups in this category roughly align with the protein level cut-offs for category 3 levels (25 mg PHE with 0.5g protein, 5 mg PHE with 0.1g protein, 75 mg PHE with 1.5g protein), so it appears very arbitrary to use separate categories. It seems like this group would more accurately be categorized as foods specifically manufactured or marketed to be low-protein… plus some random vegetables (why is this sub-group not worked into category 3?).
This section (Category 4) has been modified, further explained and a table (Table 4) added for more information.
- The descriptions (line 164, line 180-181 and the matching table 1 description for category 4) that state "based on… and if" are not clear (e.g., in the sentence "I categorize my sweaters based on colour, weight, and if they contain wool", the "and if" is not a Boolean operator for the list). Conversely, the Table 1 description for category 3 ("based on…but only if") is clearly worded.
The Table has been modified and this section has been reworded.
Discussion
- (lines 242-247): While the focus of this manuscript obviously is not the strengths and weaknesses of the various methods to administer a low phenylalanine diet, there should be some acknowledgement that certain issues are peculiar to certain counting systems. The weakness of anysystem that sets arbitrary thresholds for counting or not counting certain foods is highlighted by this section. For example, foods with phenylalanine of 23 mg /100g compared to 27 mg /100g are not all that different, but the strict (and arbitrary) cut-point of 25 mg/100g causes them to be treated differently. The European guidelines suggest the equally arbitrary cut-off of 50 mg/100g for low-protein foods. Making the statement "There seems little value in having low protein foods that must still be limited and controlled within the diet" is purely an artifact of this counting system, especially if you consider the "regular" version of that food might have 10 times the phenylalanine!
A discussion of this has been added to the introduction and the limitations section of the discussion.
Reviewer 2 Report
This paper reports the results of a consensus amongst UK metabolic dietitians on the suitability and allocation of food in a low protein diet for patients with PKU tolerating a natural protein intake of ≤10 gr/day (500 mg Phe).
The study is the result of a 4 years discussion on a list of statement, according to the Delphi methodology. The study updates the 1993 UK guidelines.
To improve the paper and the use of the results by other dietitian, also in other countries, I would suggest to expand the data in table 1. In the 2° category of table 1(Exchange free foods), vegetables containg Phe <75 mg/100 gr are reported as a group and only potatoes and vegetable crisps are excluded; it would be very useful if the authors could provide in the table the list of the vegetable that can be consumed without restriction because not only potatoes and vegetable crisps have a Phe content higher than 75 mg Phe/100gr. Otherwise the message could be misinterpreted.
In table 1, 3rd category, the allocation of foods in the sub-groups is not so clear because the authors list food in 4 groups that overlap. For example, in the list of food containing protein >0.5 gr/100 gr there are some foods (ie yoghurt or legumes) that could also be listed in the subsequent categories. Why the authors decided to allocate foods in this way? Wouldn't it be more useful to set a lower and an upper limit of each category (for example 0.1-0.5; 0.5-1, 1-1.5 and >1.5 g/100 gr)? Can the authors better clarify this choice?
Author Response
Reviewer 2
This paper reports the results of a consensus amongst UK metabolic dietitians on the suitability and allocation of food in a low protein diet for patients with PKU tolerating a natural protein intake of ≤10 gr/day (500 mg Phe).
The study is the result of a 4 years discussion on a list of statement, according to the Delphi methodology. The study updates the 1993 UK guidelines.
To improve the paper and the use of the results by other dietitian, also in other countries, I would suggest to expand the data in table 1. In the 2° category of table 1(Exchange free foods), vegetables containing Phe <75 mg/100 gr are reported as a group and only potatoes and vegetable crisps are excluded; it would be very useful if the authors could provide in the table the list of the vegetable that can be consumed without restriction because not only potatoes and vegetable crisps have a Phe content higher than 75 mg Phe/100gr. Otherwise the message could be misinterpreted.
Thank you for your comments, we have now listed exchange-free vegetables tin Table 1 and exchange vegetables (>75mg/100g) in Table 4.
In table 1, 3rd category, the allocation of foods in the sub-groups is not so clear because the authors list food in 4 groups that overlap. For example, in the list of food containing protein >0.5 gr/100 gr there are some foods (ie yoghurt or legumes) that could also be listed in the subsequent categories. Why the authors decided to allocate foods in this way? Wouldn't it be more useful to set a lower and an upper limit of each category (for example 0.1-0.5; 0.5-1, 1-1.5 and >1.5 g/100 gr)? Can the authors better clarify this choice?
The results and table have been rewritten and divided into 4 tables to better explain this. Most foods fall into the general rule of anything >0.5g/100g is an exchange food. However, there were some specific foods for which this definition was considered either too broad or too restrictive. For plant milks that can be consumed in quite large amounts, it was considered too broad and therefore a more restrictive amount of >0.1g per 100g/100ml was chosen. For cook-in/pour-over sauces that are frequently made from low protein ingredients (e.g. vegetables which have been shown in previous studies to have little impact on blood Phe levels) 0.5g/100g was considered too restrictive and so this was extended to 1g/100g. Similarly, for a few other foods that are used in smaller quantities, 0.5g/100g was considered too restrictive (mayonnaise, salad cream, dressings, sauces/pickles, spreads, cake decorations). The last group included specifically soy sauce only and this was based again on the amount used and the limitations of very few brands being <0.5g/100g.
Round 2
Reviewer 1 Report
All of my concerns have been addressed in the edited manuscript. Thank you.